# Transformer-Based CT Auto-Segmentation of Lung Metastases – A Tumor-Board Application

## Abstract

Accurate segmentation of lung nodules in computed tomography (CT) scans is challenging due to extreme class imbalance, where nodules appear sparsely among healthy tissue. Lung tumor boards often review these scans manually, a time-consuming process. This paper introduces a novel two-stage approach for lung tumor segmentation by framing the problem as anomaly detection. The method is divided into two stages, allowing each model to leverage its strengths. Stage 1 focuses on region proposal, employing a custom Deformable Detection Transformer with Focal Loss to overcome class imbalance and localize sparse tumors. In Stage 2, the predicted bounding boxes are refined into pixel-wise segmentation masks using a fine-tuned variant of Meta's Segment Anything Model (SAM) for semantic segmentation. To address the challenge of nodule sparsity and improve spatial context, a 7.5 mm Maximum Intensity Projection (MIP) is applied, aiding in the differentiation between nodules, bronchioles, and vascular structures. The model achieves a Dice coefficient of 92.4%, with 95.2% sensitivity and 93.2% precision on the LUNA16 dataset, demonstrating robust performance in real-world clinical conditions where nodule sparsity is 5%.

## 1 Introduction

Lung cancer is a leading cause of cancer-related deaths worldwide with early detection and accurate assessment being crucial for improving outcomes. Lung tumor boards, comprising oncologists, radiologists, surgeons, pathologists, and other specialists, collaboratively review complex lung cancer cases to determine the best treatment plan. Accurate segmentation of lung nodules in CT scans is essential, as it provides critical information about tumor size, location, and spread. However, tumor board evaluations typically perform segmentation manually which can be is impractical as it is time-consuming, slowing decision-making and increasing resource demands.

Clinical decision support (CDS) systems have emerged as promising tools to address these challenges, integrating clinical knowledge with patient-specific data. Although increasingly integrated into routine clinical workflows, their widespread adoption remains variable due to challenges in system integration and user engagement David C. Wyatt and de Waal (2019). Implementing a CDS system for auto-segmentation of lung metastases can enhance workflow efficiency, improve patient outcomes, and reduce costs. Despite the benefits, current models struggle with the extreme class imbalance that appears in CT data, as lung nodules appear infrequently among healthy tissue. Nodule volumes are much smaller than the overall lung volume, vary widely in size and location, and often have similar shape and density to vasculature on an axial CT slice. These challenges underscore the need for a customized architecture that addresses these limitations to ensure a reliable tool for clinicians.

## 2 Related Work

**Thoracic Computed Tomography and LUNA16 Dataset.** Thoracic Computed Tomography (CT) involves a series of 2D cross-sectional greyscale images that, when combined, form a detailed 3D representation of the patient's thorax. The LUNA16 dataset, derived from the LIDC-IDRI dataset, consists of 888 thoracic CT scans containing 1,186 annotated lung nodules. Lung nodules in the LUNA16 dataset are annotated based on the consensus of at least three out of four radiologists, with nodules larger than 3 mm considered relevant findings.

CT datasets present challenges due to the sparsity of nodules—typically only 0 to 5% of slices contain a nodule Walter et al. (2016)—and the varying voxel sizes and scan resolutions across different patients. The LUNA16 dataset serves as a critical benchmark and numerous studies have trained architectures such as CNNs, 3D-CNNs, U-Net, and V-Net on the dataset El-bana et al. (2020); Gu et al. (2018). Existing research often uses inconsistent and under-documented data processing techniques, sometimes extracting only positive tumor slices. This makes them difficult to replicate and less applicable to real-world scenarios where lung nodules sparisty is a key limitation for modelsAgnes and Anitha (2020); Bhattacharyya et al. (2023). Additionally more powerful transformer architectures have not yet been fully explored on the problem space Xiao et al. (2020).

**Handling Imbalanced Data in Deep Learning.** Imbalanced Data is challenging for models because they are optimized to minimize overall error which leads to a bias favouring the majority class over the minority class. In cases of class imbalance models are at risk of converging to a majority class classifier, which keep accuracy high but results in no detections of lung nodules.

Various mitigation strategies for class imbalance exist, including oversampling, class weighting, and focal loss. Oversampling tumor slices artificially balances the dataset by increasing the number of minority class examples, but misrepresents the prevalence of the object in real-world conditions Qu et al. (2020). Class weighting increases the loss contribution of the minority class, forcing the model to pay more attention to underrepresented cases Buda et al. (2017). However, this is also shown to increase false positives as it over-represents the minority class Chan et al. (2019; 2020). A more targeted approach, focal loss, modifies the cross-entropy loss by down-weighting well-classified examples and emphasizing hard-to-classify ones like tumors, adjusting the loss based on prediction confidence. This method avoids shortcomings of other techniques and is demonstrates improved detection for rare classes with less false positives Lin et al. (2017b).

**Transformers in Medical Computer Vision** Transformer architectures have become a strong alternative to Convolutional Neural Networks (CNNs) in medical computer vision. CNN's local receptive fields are good at capturing local features, but struggle with long-range dependencies which refer to the model's ability to understand relationships between distant parts of an image Shamshad et al. (2023). Transformers' self-attention allows the capture of these complex relationships and are crucial for allowing differentiation between nodules and vessels. Shamshad et al. (2023).

Detection Transformer (DETR) is an object detection vision transformer (ViT) architecture that uses self-attention to directly predict object locations, bypassing traditional region proposal methods Carion et al. (2020). This architecture has set new standards for finding complex objects by integrating detection as a set prediction problem Carion et al. (2020). However, DETR is known to have slow training convergence and difficulty in detecting small objects Zhu et al. (2020). Deformable-DETR is an enhancement of DETR that introduces a deformable attention mechanism. This focuses attention on a sparse set of key sampling points near the object, allowing the model to concentrate on the most informative regions rather than the entire image uniformly to improve both the efficiency and performance of the model Zhu et al. (2020). Segment Anything Model (SAM) is a foundational segmentation model designed to be promptable. It is trained on over 1 billion masks across 11 million images, and is shown to be able to efficiently transfer understanding to new data distributions through fine-tuning for advanced pixel-wise segmentation in many fieldsKirillov et al. (2023); Ma et al. (2024). MedSAM, an adaptation of SAM for medical imaging, fine-tunes SAM to focus on the specific needs of medical images, such as varying tissue densities and anatomical complexities Ma et al. (2024).

## 3 METHODOLOGY

This paper presents a novel approach to lung tumor segmentation for tumor boards by framing the task as anomaly detection. Our proposed method splits the task into two stages: Stage 1 serves as a region proposal phase to localize sparse tumors, while Stage 2 refines these bounding boxes into pixel-wise segmentation masks. This is the first method to combine various elements, including architectural components like Deformable-DETR and SAM, along with strategies such as Focal Loss and Maximum Intensity Projection (MIP), into a unified framework specifically tailored for sparse lung nodule segmentation. We developed a customized training regimen for a processed LUNA16 dataset to address severe class imbalance and focus on difficult cases. This two-stage pipeline is specifically designed to optimize segmentation performance for tumor board evaluations.

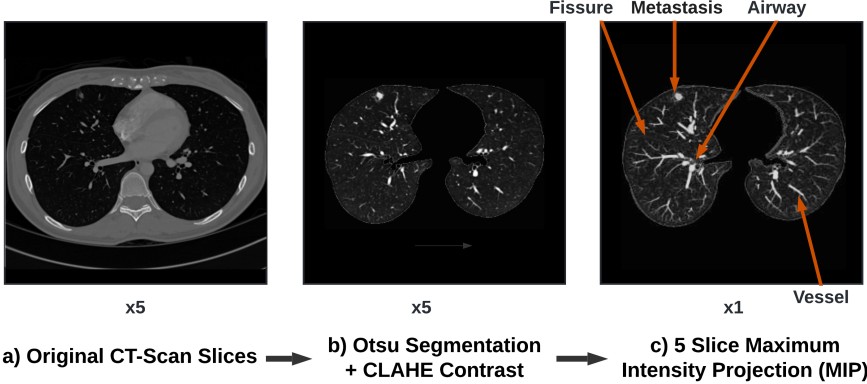

Figure 1: Data Processing Pipeline With Tumor Visible at the Top Left of Lung. a) Original CT slice, b) Post Otsu segmentation and CLAHE, c) Post 7.5mm (5 slice) Maximum Intensity Projection.

### 3.1 DATA PREPROCESSING

Our preprocessing pipeline prepares CT scan data from the LUNA16 dataset for input into Deformable-DETR. We visualize this process in Figure 1. CT data and mask annotations are loaded in MetaImage (mhd/raw) format. Since the CT datasets in LUNA16 have varying voxel sizes, including differences in cross-sectional size and slice thickness, we standardize anatomical structures by resampling the images to achieve consistent voxel spacing. Specifically, we resample the images to a target voxel size of $1 \times 1 \times 1$ mm, ensuring uniformity across the dataset. The resampling factor $R$ is calculated as shown in Equation (1), where the image is scaled accordingly to achieve the desired voxel spacing:

$$R = \frac{S}{S'} = \left[ \frac{S_x}{S'_x}, \frac{S_y}{S'_y}, \frac{S_z}{S'_z} \right] \tag{1}$$

Otsu's method is utilized to threshold and segment lung tissue from surrounding background, isolating the lung areas. This is followed by morphological operations, including connected component analysis and region erosion, to obtain clean binary masks to separate lungs from other features. Slices at the superior and inferior aspects of the dataset in the cranio-caudal direction, which provide minimal diagnostic information, are removed based on non-zero area size. This significantly reduces the model's search space from around 15 million to 5.25 million pixels per patient. After segmentation, we enhance image contrast using Contrast Limited Adaptive Histogram Equalization (CLAHE), which improves the visibility of subtle features like small nodules Sundaram et al. (2011). This process is illustrated by the leftmost arrow of Figure 1.

Maximum Intensity Projection (MIP) enhances nodule visibility by combining adjacent CT slices into a single 2D image by projecting the highest attenuation voxel from a 3D volume onto a 2D image, preserving crucial 3D spatial information. Cody (2002). Widely used by radiologists, MIP helps distinguish nodules which generally appear as compact blobs, whereas vessels are elongated tube-like structures. This method is shown to be extremely effective in detecting small pulmonary nodules between 3 mm and 10 mm while also reducing false positives Gruden et al. (2002); Zheng et al. (2019). This process can be mathematically described by Equation (2), where the highest intensity voxel along the z-axis is selected for each (x, y) coordinate producing a 2D image that highlights the most dense features. A slab thickness of 7.5mm was found to be a suitable compromise between differentiating vessel and nodule shape and limiting overlap between structures. This process is illustrated by the rightmost arrow of Figure 1.

$$I_{\text{MIP}}(x, y) = \max_z \{ I(x, y, z) \} \tag{2}$$

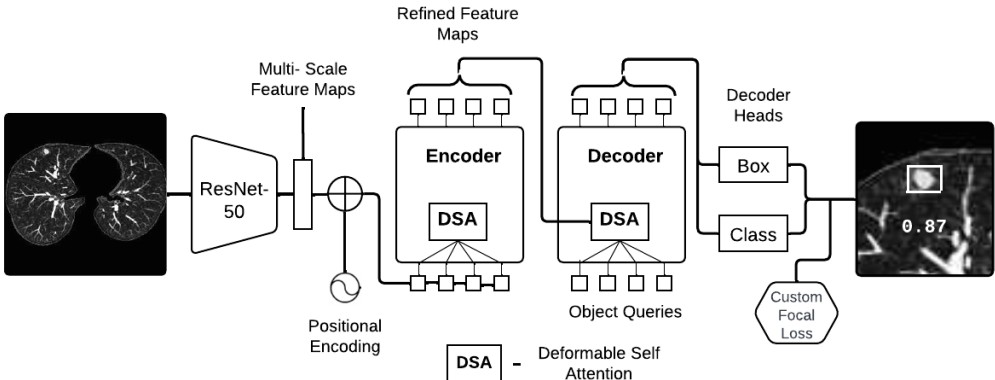

Figure 2: Stage 1 - Region Proposal with Custom Deformable-Detection Transformer Architecture and Focal Loss

## 3.2 DATASET

The final processed dataset consists of 9,676 CT scan slices, each with a 7.5mm Maximum Intensity Projection (MIP) applied. Among these, 1,226 images are annotated with nodules, while the remaining 8,450 images contain healthy tissue. The dataset was split into 70% for training, 20% for validation, and 10% for testing prior to any augmentation to avoid data contamination and ensure rigorous evaluation. In the training and validation sets, 12.7% of the images contained a lung nodule. To better mimic real-world conditions, the test set had a reduced lung nodule rate of 5%, contrasting with the higher rate used during training. This elevated rate in training was necessary to strike a balance between realism and model performance, as lower rates resulted in a dataset too sparse for effective training. Post-split, a set of data augmentations was applied to the training set to increase the dataset's size and variability. These include horizontal and vertical flips, rotations between -15° and +15°, brightness adjustments within -15% to +15%, and Gaussian noise (0.001 to 0.18% SD) simulated typical CT scan sensor noise.

## 3.3 STAGE 1: REGION PROPOSAL

Stage 1 of our approach focuses on generating region proposals to localize potential tumor candidates within lung CT scans. Detection Transformer (DETR) was chosen as the architecture for proposing bounding boxes due to its strong performance in complex object detection tasks, particularly since it eliminates the need for anchor boxes and ability to handle diverse object sizes and shapes within the same image. The Deformable variant, introduced by Zhu et al. Zhu et al. (2020), was selected for its spatially adaptive and computationally efficient attention mechanism. Initial experimentation with DETR yielded a sensitivity of 42% after 20 epochs, specifically struggling with tumors under 10mm. Switching to Deformable-DETR improved sensitivity to over 90% across all tumor sizes after just 8 epochs. With 74% of tumors in the LUNA16 dataset measuring 3-10mm, the deformable attention variant was chosen for tumor detection.

Figure 2 shows our custom Deformable-DETR architecture which is trained from scratch for detection of sparse lung nodules, evaluated using IoU metrics to inform the customized loss discussed in section 3.6. MIP images are processed through a ResNet-50 CNN backbone to extract multi-scale feature maps, capturing both low-level textures and high-level semantic features. These are augmented with 2D sine-cosine positional encodings to preserve spatial context and then fed to the encoder. The encoder uses Deformable Self-Attention (DSA) layers to refine multi-scale feature maps, attending to a sparse set of learnable sampling points around each nodule. The computational complexity of self-attention is $O(H^2W^2C)$, where $H = 256$, $W = 256$ represent the feature map height and width in pixels, and $C = 1$ represents the number of channels for grayscale images. The encoder also integrates a multi-scale attention mechanism to process information at different feature scales, enhancing the model's ability to detect nodules of varying sizes. The encoder outputs refined

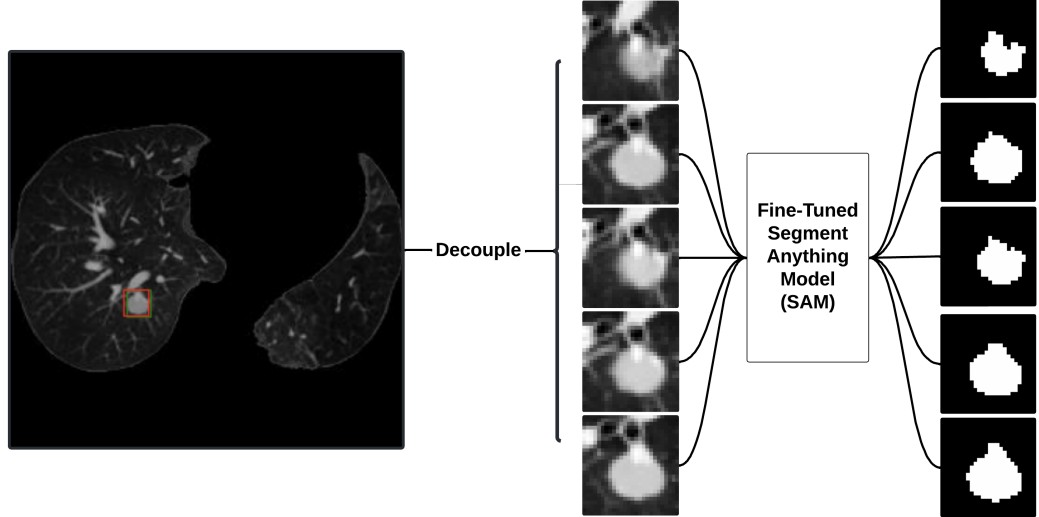

Figure 3: Stage 2 Auto-Segmentation model. Bounding boxes from Stage 1 are split into individual CT slices, cropped into patches, and passed into the model for pixel-wise segmentation.

multi-scale feature maps enriched with context-aware representations. The decoder stage takes these and combines them with object queries, a learnable set of positional embeddings representing potential nodules. The decoder's outputs are finally processed by the Bounding Box Regression and Classification Heads to convert raw features into predicted nodule coordinates and classification probabilities, ensuring precise localization and accurate differentiation of nodules.

### 3.4 STAGE 2: AUTO-SEGMENTATION MODEL

In the second stage, we utilize our auto-segmentation model, a fine-tuned adaptation of the Segment Anything Model (SAM) with initial weights imported from MedSAM, visualized in Figure 3. This model is used for precise segmentation by splitting the MIP images back into their individual CT slices and generating pixel-wise segmentation masks for each slice. The bounding boxes produced by Deformable-DETR are integrated into SAM's segmentation process through the prompt encoder, which is designed to handle various input types such as boxes, points, and masks. These bounding box embeddings act as attention cues, encoded into high-level features that instruct our auto-segmentation model to focus on specific areas. The goal is to improve accuracy by guiding the segmentation algorithm to specific regions of the images that are more likely to contain nodules, while still retaining the entire image for context.

### 3.5 TRAINING

Deformable-DETR was trained and evaluated in a Google Colab environment using an L4 GPU, providing sufficient power for high-resolution 3D CT scans. The model was trained for 15 epochs using the AdamW optimizer, with learning rates of 1e-4 for the main parameters and 1e-5 for the backbone, and a weight decay of 1e-4 to mitigate overfitting. A Step Learning Rate Scheduler adjusted the learning rate dynamically, reducing it by a factor of 10 every 10 epochs. The model used a batch size of 6 with mixed precision (16-bit) for improved speed and efficiency. Gradient clipping was set at 0.1, and gradients were accumulated over 6 batches to ensure stable learning.

In Stage 2 our auto-segmentation model was trained by loading model checkpoints from MedSAM Ma et al. (2024) and fine-tuning SAM's architecture using a dataset comprised of processed LUNA16 images and their associated ground truth pixel masks. Dice-CrossEntropy Loss was selected to optimize the overlap between predicted and ground truth masks while mitigating class imbalance with Cross-Entropy Loss. The model was trained using the Adam optimizer specifically on the mask decoder.

Figure 4: Stage 1 Region Proposals on six CT slices from the LUNA16 test dataset, with green boxes indicating ground truth and red boxes showing model predictions.

### 3.6 FOCAL LOSS FOR CLASSIFICATION

To handle the significant class imbalance in the LUNA16 dataset, we customize the DETR loss function to incorporate focal loss. By adding a modulating factor, focal loss down-weights well-classified samples and emphasizes hard-to-classify samples, assisting in the detection of rare nodule instances Lin et al. (2017a). The focal loss function is defined in Equation (3) Lin et al. (2017a):

$$FL(p_t) = -\alpha_t(1 - p_t)^\gamma \log(p_t), \tag{3}$$

where $p_t$ is the predicted probability of the correct class, $\alpha_t$ balances positive and negative examples, and $\gamma$ adjusts focus towards challenging samples.

## 4 RESULTS

Table 1: Performance Metrics for Region Proposal (Stage 1) and Auto-Segmentation (Stage 2) for Sparse Lung Tumor Detection

| Metric | F1/Dice | Precision | Sensitivity |
|---|---|---|---|
| **Stage 1: Region Proposal** | | | |
| F1 Score | 94.2% | – | – |
| Average @ IoU 0.5 (All Areas) | – | 93.3% | 95.2% |
| Average @ IoU 0.5 (Small Areas) | – | 78.4% | 83.3% |
| Average @ IoU 0.5 (Medium Areas) | – | 96.7% | 97.0% |
| Average @ IoU 0.5 (Large Areas) | – | 97.8% | 99.2% |
| **Stage 2: Auto-Segmentation** | | | |
| Dice Coefficient | 92.4% | – | – |

This section evaluates the performance of the proposed two-stage architecture on the LUNA16 test dataset with a focus on key metrics. Table 1 summarizes the performance metrics for Stage 1 (Region

Proposal) and Stage 2 (Auto-Segmentation) on the LUNA16 test dataset, with nodules categorized by size: small (up to 7 mm), medium (7-15 mm), and large (over 15 mm). Precision reflects the proportion of correctly identified nodules among all predicted nodules, while sensitivity measures the proportion of actual nodules that were successfully detected. The F1 score combines these metrics for balanced accuracy, and the Dice Coefficient evaluates the overlap between predicted and actual segmentation masks.

Through hyperparameter tuning, we found that setting $\gamma = 2$ and $\alpha_t = 0.25$ provided the best balance between precision and recall. These values helped the model focus on harder-to-classify nodules, reducing false positives and negatives. The value of $\gamma$ emphasized challenging examples, while $\alpha_t$ balanced positive and negative samples.

Stage 1 proposed regions achieve strong precision and recall across most tumor sizes, even with a test nodule sparsity of 5%. For medium and large tumors, the model maintains high precision (96.7% and 97.8%) and recall (97.0% and 99.2%). Stage 2 achieves a Dice Coefficient of 92.4%, closely matching ground truth masks. The accuracy of this stage depends on the quality of predicted bounding boxes, with only a slight drop in accuracy from Stage 1 (94.2% F1 score), demonstrating the robustness of the auto-segmentation process with a high retention of approximately 98%.

Figure 4 provides Stage 1 visualizations of proposed regions on six CT slices, showing green boxes for ground truth and red boxes for predictions. Despite the complexity of vascular structures and bronchioles, the model's predictions align closely with the ground truth in all slices.

## 5 DISCUSSION

Table 2: Comparison of Lung Nodule Segmentation Models on Dice Coefficient, Sensitivity, and Specificity

| Author | Architecture | Dice Coefficient (%) | Sensitivity (%) | Specificity (%) |
|---|---|---|---|---|
| Agnes et al. Agnes and Anitha (2020) | MRUNet-3D | 89.0 | 94.8 | 84.2 |
| Bhattacharyya et al. Bhattacharyya et al. (2023) | DB-NET | 88.9 | 90.2 | 77.9 |
| Song et al. Song et al. (2023) | ConvLSTM | 84.0 | 87.8 | 81.5 |
| Ma et al. Ma et al. (2023) | SW-UNet | 84.0 | 82.0 | 89.0 |
| Annavarapu et al. Annavarapu et al. (2023) | Bi-FPN | 82.8 | 92.2 | 78.9 |
| Cao et al. Cao et al. (2020) | DB-ResNet | 82.7 | 89.4 | 79.6 |
| Tyagi et al. Tyagi and Talbar (2022) | CSE-GAN | 80.7 | 85.5 | 77.5 |
| Wang et al. Wang et al. (2017) | MV-DCNN | 77.9 | 87.0 | 77.3 |
| Sun et al. Sun et al. (2017) | MCROI-CNN | 77.0 | 85.4 | 73.5 |
| **Our Proposed Method** | **DETR-SAM** | **92.4** | **95.2** | **93.3** |

Our two-stage approach outperforms many established models for lung nodule segmentation, particularly those based on traditional CNN and U-Net architectures. Models like U-Net, V-Net, and MRUNet-3D often struggle with small sparse nodules. Architectures like MRUNet-3D Agnes and Anitha (2020) and DB-Net Bhattacharyya et al. (2023) address these challenges through multi-scale feature extraction and deeper networks. 3D-MSViT, a hybrid CNN-transformer approach used by Mkindu et al. Mkindu et al. (2023), achieved strong specificity (97.8%) and high sensitivity but was primarily focused on detection, highlighting the limitations of hybrid models in segmentation. Similarly, SW-UNet Ma et al. (2023), which integrates a sliding window transformer with a CNN, showed improved segmentation accuracy, yet its performance suggests that fully transformer-based models may better handle the complexities of precise lung nodule segmentation.

While existing methods like 3D-MSViT Mkindu et al. (2023) and SW-UNet Ma et al. (2023) have integrated transformers in a hybrid manner, our approach aims to fully exploit the benefits of transformers by custom-training models like Deformable-DETR and SAM. These models enable both direct detection and precise pixel-wise segmentation, addressing the complexities that hybrid approaches often miss Carion et al. (2020); Zhu et al. (2020). This approach allows for higher capacity models to better decipher complex structures potentially setting a new standard for accuracy in lung nodule segmentation.

An important point of differentiation is transparency regarding class sparsity. Many existing methods use varying data processing techniques that may emphasize positive tumor slices, which can complicate replication and reduce applicability to real-world scenarios where class imbalance is significant Cao et al. (2020). In contrast, our approach preserves the natural class imbalance found in clinical settings, ensuring that both positive and negative samples are accounted for in the model design and evaluation. When tested on a dataset with higher nodule sparsity, our model demonstrated superior performance, particularly in detecting small tumors, a crucial factor in early diagnosis.

In our analysis, we discovered that SAM, when deployed independently, struggled with class imbalance, achieving only a 45% Dice Coefficient and low overall accuracy. This highlighted a critical gap: SAM's capabilities were not well-suited for the initial detection phase. To bridge this gap, we integrated Deformable-DETR with a custom loss function as the first stage. This model excelled at localizing nodule regions, narrowing SAM's task to precise segmentation rather than detection and significantly improving the performance. This division of labor between Deformable-DETR and SAM allowed each model to operate within its strengths.

Stage 1 shows relatively lower precision and recall for small nodules (up to 7 mm in diameter), reflecting the inherent challenges of detecting small nodules due to their lower contrast in CT scans. Notably, the prevalence of malignancy in nodules smaller than 6 mm is very low, ranging between 0 and 1%, and guidelines from the European Respiratory Society now suggest a threshold of 6 mm for follow-up consideration due to the low malignancy risk associated with these small nodules Larici et al. (2017).

## 6 CONCLUSION

This study introduces a two-stage framework for lung nodule segmentation. Stage 1 focuses on region proposal, achieving an F1 score of 94.2% for accurate bounding box predictions. Stage 2 refines these regions into pixel-wise segmentation masks, resulting in a 92.4% Dice coefficient on the LUNA16 dataset. This framework demonstrates significant improvements and successfully integrates modern transformer architectures. Looking forward, future research will focus on validating the model across diverse clinical datasets to enhance generalizability and refining its detection capabilities for small tumors, addressing a key challenge in early lung cancer diagnosis.

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
