# OpenReview forum: "Transformer-Based CT Anomaly Detection and Auto-Segmentation of Sparse Lung Nodules"
_ICLR.cc/2025/Conference — Submitted to ICLR 2025_

### Official Review · Reviewer_tPtV · 2024-10-17

**Soundness:** 2
**Presentation:** 3
**Contribution:** 1
**Rating:** 1
**Confidence:** 5

**Summary:**

This manuscript presents a novel two-stage approach for automating lung nodule segmentation using transformer models. In the data preprocessing phase, Maximum Intensity Projection (MIP) enhances spatial features, helping to distinguish nodules from bronchioles and vessels in CT images. Next, region proposal bounding boxes are generated using the Deformable-DETR model. In Stage 2, these bounding boxes are processed by the SAM model to achieve pixel-level segmentation. To address class imbalance within the dataset, focal loss is incorporated into the original DETR loss function. The results demonstrate superior performance compared to state-of-the-art (SOTA) methods.

**Strengths:**

In preprocessing, maximum intensity projection is applied across slices to enhance visibility. The two-stage framework, combining DETR and SAM models, offers a straightforward approach. Additionally, the common issue of class imbalance in medical datasets is addressed. As a result, segmentation performance is significantly improved. The paper is well organized.

**Weaknesses:**

This manuscript lacks novel insights, as the deep learning models used in each stage are well-established, and focal loss is widely applied across various domains. Additionally, the ROI-based segmentation approach is considered somewhat conventional. There is no ablation studies on original SAM performance and some critical models to compare or discuss are missing.

**Questions:**

1. What is the technical insight of this work? The concept of ROI-based segmentation is not new, and both DETR and SAM are well-established models. As a result, the framework appears to lack novelty, which is a critical concern.

2. There are several existing studies on lung nodule segmentation in CT images. For instance, the IEEE TMI paper, "Closing the Gap between Deep Neural Network Modeling and Biomedical Decision-Making Metrics in Segmentation via Adaptive Loss Functions," addresses not only lung segmentation but also class imbalance. It would be beneficial for the authors to compare or discuss their work in relation to such prior studies.

3. Does the model function in an end-to-end learning manner? If it does or does not, the authors should provide a discussion on the merits and limitations of the learning method used in this framework.

4. Which stage of the framework provides the most significant performance improvement?

5. Additionally, what is the baseline performance of the original SAM model pretrained with MedSAM? Are the datasets used in MedSAM aligned with the LUNA dataset, and how does this impact model performance?

---

### Official Review · Reviewer_dyCf · 2024-10-29

**Soundness:** 2
**Presentation:** 1
**Contribution:** 2
**Rating:** 3
**Confidence:** 5

**Summary:**

This paper proposes a two-stage pipeline for lung nodule segmentation in CT scans, designed to support lung tumor boards by enhancing segmentation accuracy and efficiency. The first stage employs a custom Deformable Detection Transformer (DETR) architecture to detect sparse lung tumors, leveraging deformable attention to improve sensitivity to small nodules. The second stage utilizes a fine-tuned Segment Anything Model (SAM), enhanced with medical imaging capabilities (MedSAM), to refine the bounding boxes into pixel-level segmentation masks, ensuring precision in differentiating nodules from surrounding anatomy.

To address the class imbalance in CT data - where lung nodules are rare compared to healthy tissue - the framework incorporates focal loss, reducing model bias towards non-tumor areas and enhancing detection accuracy for hard-to-detect nodules. Achieving a 94.2% F1 score for bounding box prediction and a 92.4% Dice coefficient in segmentation accuracy, this pipeline demonstrates strong potential to improve clinical workflows, enhance tumor board decision-making, and contribute to better patient outcomes by streamlining nodule detection in a clinical setting.

**Strengths:**

The paper introduces a novel two-stage approach for lung tumor segmentation by framing the task as anomaly detection, addressing the challenges of sparse nodule identification in CT scans. This innovative structure uses a Deformable Detection Transformer (DETR) for region proposals and a fine-tuned Segment Anything Model (SAM) for precise segmentation, effectively handling class imbalance and complex image features.

Additionally, the paper’s clear, logical structure and well-explained methodology make complex concepts accessible. Detailed quantitative results further highlight the framework's effectiveness, making it a valuable and readable contribution to medical imaging and clinical decision support.

**Weaknesses:**

1. The paper’s experimental section lacks depth, with insufficient analysis to thoroughly validate the proposed method.

2. There is no ablation study provided, which limits insight into how each component - such as the use of Deformable Detection Transformer (DETR), the fine-tuned Segment Anything Model (SAM), and the customized focal loss - contributes to overall performance. Without this breakdown, it’s difficult to assess which aspects of the framework are most effective.

3. The paper relies solely on quantitative evaluation, omitting any qualitative assessment, such as visual comparisons among different methods, which could provide a clearer understanding of the model's segmentation accuracy and real-world applicability.

4. The presentation of results is also weak, with layout issues that detract from readability and professionalism. For instance, Table 2 extends beyond the page margin, rendering the data difficult to interpret. Additionally, there are inconsistencies and errors in in-text citations, which may confuse readers and hinder the paper’s credibility. These issues in presentation and citation detract from the paper's overall clarity and polish, suggesting the need for more careful formatting and editing. There are also several grammatical errors, which make the paper somewhat challenging to read.

**Questions:**

- Why is there only quantitative evaluation/comparison provided in the manuscript? Could you provide some qualitative results, such as visual examples of segmentation outputs, to illustrate the model’s performance?

- Can you elaborate on how the class imbalance was handled during training? Were any additional strategies (besides focal loss) considered or tested to further address this issue?

---

### Official Review · Reviewer_rtSW · 2024-11-04

**Soundness:** 2
**Presentation:** 2
**Contribution:** 2
**Rating:** 3
**Confidence:** 5

**Summary:**

This study is to propose a two-stage approach for lung tumor segmentation by anomaly detection including stage 1 of region proposal with deformable detection transformer with focal loss, and stage 2 with fine-tuned SAM. This study is notable for its use of the MIP (Maximum Intensity Projection) method to address issues related to nodule sparsity and spatial context. This approach is also frequently employed by radiologists. However, the primary concern with this paper is that all preprocessing and modeling steps are performed in 2D. When lung segmentation is conducted in 2D, it may be challenging to differentiate diseased lungs or lung cancers that are close to the thoracic wall. Additionally, for nodules with subsolid or GGO characteristics, visibility might be reduced in thicker MIP slices, suggesting that these types should be evaluated separately. Despite achieving better results than previous models, the study lacks an analysis of subclasses or an ablation study, and falls short in terms of technical novelty.

**Strengths:**

This study is to propose a two-stage approach for lung tumor segmentation by anomaly detection including stage 1 of region proposal with deformable detection transformer with focal loss, and stage 2 with fine-tuned SAM. This study is notable for its use of the MIP (Maximum Intensity Projection) method to address issues related to nodule sparsity and spatial context. This approach is also frequently employed by radiologists.

**Weaknesses:**

The primary concern with this paper is that all preprocessing and modeling steps are performed in 2D. When lung segmentation is conducted in 2D, it may be challenging to differentiate diseased lungs or lung cancers that are close to the thoracic wall. Additionally, for nodules with subsolid or GGO characteristics, visibility might be reduced in thicker MIP slices, suggesting that these types should be evaluated separately. Despite achieving better results than previous models, the study lacks an analysis of subclasses or an ablation study, and falls short in terms of technical novelty.

**Questions:**

Preprocessing is performed with 1 mm isocubic resolution, yet the method for generating 7.5 mm MIP using five slices in Figure 1-c needs clarification.

The authors should analyze the histogram of nodules under 10 mm from the LIDC dataset and include these results and discussions in the paper.

The paper lacks external validation, which raises concerns about the generalizability of the findings. A discussion on this limitation is recommended.

In Figure 1-b, lung segmentation is reportedly performed using Otsu segmentation; however, accuracy metrics such as DSC should be presented.

**Details Of Ethics Concerns:**

Public dataset

---

### Official Review · Reviewer_LSry · 2024-11-04

**Soundness:** 2
**Presentation:** 2
**Contribution:** 1
**Rating:** 3
**Confidence:** 5

**Summary:**

This paper introduces a two-stage approach for lung tumor segmentation in LUNA16 CT scans tackling the challenge of nodule sparsity and class imbalance. Stage 1 uses a custom Deformable Detection Transformer with Focal Loss for region proposals while Stage 2 refines these into segmentation masks with a fine-tuned variant of segment anything model. Maximum intensity projection enhances spatial context, improving differentiation between nodules, bronchioles, and vessels. The model achieves a dice coefficient of 92.4%, with high sensitivity and precision.

**Strengths:**

I like the idea of using maximum intensity projection to help neural networks clearly distinguish between nodules, bronchioles, and vessels.

**Weaknesses:**

- proposed framework brings together known architectural components like Deformable-DETR and SAM. I think the paper fall short in showing how these individual components are fundamentally innovated upon rather than simply integrated. Without clear evidence of novel adaptations or improvements to each component, the approach might seem like an assemblage of established methods rather than a groundbreaking technique
- although model achieves strong results on a modified version of the LUNA16 dataset, its robustness in diverse real-world settings is uncertain. Validating the approach on additional lung datasets could reinforce its practical impact and help mitigate concerns of overfitting to a single dataset
- paper introduces encoding and decoding processes using Deformable-DETR and SAM yet it lacks a detailed mathematical explanation of how refined feature maps are encoded and subsequently reconstructed. Without mathematical formulations and a proof of concept, it remains unclear how effectively the feature maps capture and retain essential characteristics through the two stages

**Questions:**

n/a

---

### Meta-Review · Area_Chair_KmDz · 2024-12-22

**Metareview:**

This work presents a two-stage approach for lung tumor segmentation by framing the problem as anomaly detection.

**Additional Comments On Reviewer Discussion:**

This work has four reviewers, but all of them are negative about accepting this work. Their scores are 3, 1, 3, and 3. Moreover, the authors do not provide any rebuttal. In this regard, this work can not be accepted.

---

### Decision · Program_Chairs · 2025-01-22

Reject